# Effect of Konjac Glucomannan (KGM) on the Reconstitution of the Dermal Environment against UVB-Induced Condition

**DOI:** 10.3390/nu12092779

**Published:** 2020-09-11

**Authors:** Kyung Ho Choi, Sung Tae Kim, Bum Ho Bin, Phil June Park

**Affiliations:** 1Department of Applied Biology, Ajou University, 206 World Cup-ro, Yeongtong-gu, Suwon-si, Gyeonggi-do 16499, Korea; c457854@naver.com; 2Department of Pharmaceutical Engineering, Inje University, Gimhae-si, Gyeongsangnam-do 50834, Korea; stkim@inje.ac.kr; 3Department of Nanoscience and Engineering, Inje University, Gimhae-si, Gyeongsangnam-do 50834, Korea; 4AMOREPACIFIC R&D Center, 1920 Yonggu-daero, Giheung-gu, Yongin-si, Gyeonggi-do 17074, Korea

**Keywords:** konjac glucomannan, ultraviolet B, human epidermal primary melanocytes, human embryonic fibroblasts, senescence

## Abstract

Skin layers serve as a barrier against unexpected critical changes in the body due to environmental factors. Excessive ultraviolet (UV) B exposure increases the levels of age-related factors, leading to senescent cells and damaged skin tissues. Widely used as a dietary supplement, konjac (*Amorphophallus konjac*) glucomannan (KGM) has shown skin regeneration potential in patch or sheet form with anti-inflammatory or immunosuppressive effects. However, the ability of KGM to reconstitute senescent/damaged skin following UV radiation has not been explored. Here, we demonstrate that KGM alleviates skin damage by increasing the proportion of young cell populations in UVB-exposed senescent human epidermal primary melanocytes. Young cell numbers increased depending on KGM dosage, but the senescent cells were not removed. Real-time quantitative polymerase chain reaction (RT-qPCR) and Western blot analysis showed that mRNA and protein levels of age- and pigmentation-related factors decreased in a manner dependent on the rate at which new cells were generated. Moreover, an analysis of mRNA and protein levels indicated that KGM facilitated youth by increasing cell proliferation in UVB-damaged human fibroblasts. Thus, KGM is a highly effective natural agent for maintaining skin homeostasis by promoting the reconstitution of the dermal environment against UVB-induced acute senescence or skin damage.

## 1. Introduction

The human skin, comprising almost 16% of the body, maintains a certain level of physiological and biological function (homeostasis) to ensure that the various skin cells beyond the dermis layer are preserved to form a solid barrier [1,2,3]. In particular, the epidermis layer of the skin is a critical component for maintaining the physiological functions of the body in the face of various external environmental changes as well as internal factors [4,5,6,7].

Among the diverse external factors that cause changes in the skin, ultraviolet radiation (UVR) is a well-known major contributor to human skin aging acceleration by inducing DNA damage (photo- aging) or oncogene activation [8,9,10]. UVR is classified according to wavelength as UVA (315–400 nm), UVB (280–315 nm), and UVC (<280 nm) rays, each of which has distinctive skin penetration depth due to their respective retention energies [11,12]. A recent study demonstrated that exposure to UVR, especially UVB, led to pigment accumulation and senescence in human epidermal primary melanocytes (HEMns) [13]. In addition, UVB-induced changes in p53 gene expression were identified as a major contributor to pigmentation [13]. Skin aging caused by long-term UVB exposure cannot be easily reverted to normal homeostatic conditions, and the accumulated damage can easily activate aging-related factors even upon exposure to weak stimuli. Thus, accelerated aging-related factors can increase the chance of forming abnormal cell clusters, leading to critical pathological conditions such as melanoma [14,15,16].

Konjac (*Amorphophallus konjac*) is a perennial plant that grows in the subtropical regions of South East Asia and Africa [17]. Konjac glucomannan (KGM), a heteropolysaccharide produced from the tubers of *A. konjac*, is widely used in various foods and medicines owing to its unique physical and chemical properties [18,19,20,21]. KGM is composed of β-1,4-linked d-glucose and d-mannose at a 1:1.6 molar ratio and has a characteristic randomly acetylated structure at the C-6 position per approximately every 19 sugar units [22,23,24]. Due to its water absorptivity, safety, and stability, KGM has recently been demonstrated to be a valuable source of hydrocolloidal dietary fiber and has been proposed as a useful nutritional supplement for the treatment and prevention of obesity, diabetes, and related symptoms [25,26,27]. In addition to its broad use in food and medicine, the applications and popularity of KGM have been increasing with recognition of its inherent biocompatibility, biodegradability, renewability, and nontoxicity [28,29,30,31]. Moreover, studies regarding KGM in patch form have demonstrated its strong cell regeneration capabilities, suggesting its potential in the treatment of various dermatological conditions, including wound healing [32,33].

However, the ability of KGM to reconstitute the dermal environment against UVR-induced senescence/damage and the underlying mechanism of action remain unclear. Therefore, the aim of the present study was to investigate the effect of KGM on HEMns and human embryonic fibroblasts (HEFs) after UVB exposure. We focused on the changes in the extent of UVB-induced senescence/damage to the skin cells, after two weeks of culture in KGM. Moreover, we explored whether hyperpigmentation is regulated by KGM in UVB-induced acute senescence in HEMns. Furthermore, we evaluated the relationship between physiological changes and related factors by measuring mRNA and protein expression levels. Our results highlight KGM as a novel natural anti- aging agent in addition to its currently well-known health and nutritional benefits.

## 2. Materials and Methods

### 2.1. Chemicals

Purified konjac glucomannan (KGM) powder (95% content) was purchased from Konson Konjac Co. (Wuhan, Hubei, China).

### 2.2. Cell Culture and UVB Irradiation

Neonatal foreskin-derived HEMns and HEFs were purchased from Life Technologies (Carlsbad, CA, USA). HEMns were cultured with Medium 254 (M254; Life Technologies) supplemented with human melanocyte growth supplement (Life Technologies), and HEFs were cultured in Dulbecco’s modified Eagle medium (Thermo Fisher Scientific, Waltham, MA, USA) with 5% fetal bovine serum (Thermo Fisher Scientific) in a humidified incubator under 5% CO_2_ conditions. Cells at passages 2–5 were used in subsequent experiments.

To accelerate the senescence or damage of HEMns and HEFs, UVB irradiation was applied as previously described [13]. Briefly, plated cells were rinsed twice with phosphate-buffered saline (PBS) containing calcium and magnesium (Corning Life Science, Glendale, AZ, USA), and 1 mL of PBS was added to the cells to prevent them from drying out during UV exposure. Cells were irradiated with UVB (20 mJ/cm^2^) twice at 24-h intervals using a Bio-Sun irradiator (Vilber Lourmat, Marne-la-Vallée, France). Then, the PBS was removed immediately and replaced every 2 days with growth medium with or without KGM. The cells were then cultured for 2 more weeks. Control cells were sustained in the same culture conditions without UVB exposure.

### 2.3. Cell Viability Assay

The viability of HEMns after KGM treatment was measured using the EZ-Cytox assay kit (Daeil Lab Service, Seoul, Korea) according to the manufacturer’s instructions. HEMns were cultured for 1 day in 96-well plates and treated with various concentrations (0.01–100 μg/mL) of KGM for 24 h and 72 h, respectively. EZ-Cytox solution (10 μL) was then added to each well, followed by incubation at 37 °C for 2 h. Absorbance at 450 nm was measured with a spectrophotometer (Synergy H2; BioTek, VT, USA). The experiments were performed in triplicate, and the data were presented as the percentage of absorbance values.

### 2.4. SA-β-Galactosidase Assay and Staining

The degree of senescence was determined using the mammalian β-Gal assay kit (Thermo Fisher Scientific, Waltham, MA, USA) according to the manufacturer’s instructions. Following 2 weeks of cultivation, cells were washed twice with cold PBS and collected for protein extraction using M-PER mammalian extraction reagent (Thermo Fisher Scientific, Waltham, MA, USA). An equivalent amount of β-Gal assay reagent was added to 50 μL of the supernatant obtained after centrifugation at 15,000 rpm for 15 min. Following incubation for 30 min at 37 °C, absorbance was measured at 405 nm using a Synergy H2 microplate reader (BioTek, Winoosk, VT, USA). The experiment was performed in triplicate, and the data are presented as the percentage of absorbance values.

Microscopy images of cells were stained using the senescence cell histochemical staining kit (#CS0030; Sigma-Aldrich, St. Louis, MO, USA) according to the manufacturer’s instructions. Following 1 week of cultivation, cells were washed twice with cold PBS, fixed with 1 mL 4% formaldehyde solution (Wako, Kyoto, Japan) and incubated for 2 h at room temperature. To visualize the physiological changes, images and number of β SA-β-Gal stained cells in a given area were obtained using an optical microscope, IX-75 (Olympus, Tokyo, Japan).

### 2.5. RNA Extraction and Real-Time Quantitative Polymerase Chain Reaction (RT-qPCR)

Total RNA was extracted using the ReliaPrep™ RNA Cell Miniprep System (Promega, Madison, WI, USA) according to the manufacturer’s instructions, and cDNA was synthesized from approximately 1 μg total RNA using the RevertAid First Strand cDNA Synthesis Kit (Thermo Fisher Scientific, Waltham, MA, USA). The cDNA was used as a template for RT-qPCR analysis on a 7500 Fast Real-Time PCR System (Life Technologies) using the following TaqMan probes: Tyrosinase-related protein 1 (TRP1; Hs00167051_m1), tyrosinase-related protein 2 (DCT; Hs01098278_m1), tyrosinase (TYR; Hs00165976_m1), interleukin 1 beta (IL-1β; Hs01555410_m1), cyclin dependent kinase inhibitor 2A (p16^INK4A^; Hs00923894_m1), and glyceraldehyde 3-phosphate dehydrogenase (GAPDH; #4352339E). Data were obtained from three independent experiments and are presented as the fold change relative to GAPDH level in the sample.

### 2.6. Western Blot Analysis

Proteins were extracted from each sample, at 5 days and 2 weeks after UVB exposure. Total protein (15 μg) was separated by sodium dodecyl sulfate–polyacrylamide gel electrophoresis and blotted with anti-H2A histone family member X (H2AX), anti-γ-H2AX, anti-p53, anti-phospho-p53 (Ser15), anti-acetyl-p53 (Lys382), and anti-p21CIP1 antibodies (Cell Signaling Technology, Danvers, MA, USA), and with anti-TYR, anti-TYRP1, anti-TYRP2, and anti-GAPDH antibodies (Santa Cruz Biotechnology, Dallas, TX, USA), respectively.

### 2.7. Statistical Analysis

All data are presented as the mean ± SD. Two-tailed Student’s *t*-tests were used to analyze the differences between pairs of groups, and differences among multiple groups were analyzed using one-way ANOVA. The threshold for statistical significance was set at *p* < 0.05.

## 3. Results

### 3.1. UVB-Induced Senescence in Human Epidermal Melanocytes Is Alleviated by KGM Treatment

KGM used in these experiments was extracted from the roots of *A. konjac* (Figure 1). The specifications of KGM as supplied by the manufacturer are presented in Table 1. Since these specifications indicated the presence of minimal levels of ash or heavy metals such as lead and arsenic (Table 1), we performed cell viability assays to obtain the appropriate KGM concentrations for culturing HEMns. The results indicated that KGM had no notable changes in cell survival at 1, 2, 5, 10, and 20 μg/mL (Appendix A), and these concentrations were therefore used in subsequent experiments.

To induce senescence, cells were exposed to UVB (20 mJ/cm^2^) twice and cultured with KGM containing media for two weeks (Figure 2A) as described previously [13]. Microscopy analysis of morphological changes in confluent cells revealed densely packed and healthy cells in unexposed normal control (CTL) cells (Figure 2B, upper left panel). In contrast, the UVB group exhibited spatial separation marked by cell death, and surviving cells displayed morphological abnormalities characterized by enlarged sizes and numerous dendrites (Figure 2B, upper right panel). These morphological changes were consistent with melanocyte senescence reported in a previous study [13]. Notably, culturing of HEMns with 20 μg/mL KGM for two weeks compensated for the cell death-mediated spacing (Figure 2B, lower right panel). Specifically, the population of the UVB- exposed HEMns gradually increased in the presence of KGM in a dose-dependent manner (Figure 2B, lower panel), suggesting that KGM induces cell proliferation in UVB-exposed HEMn.

### 3.2. KGM Promotes the Growth of UVB-Induced Senescent Human Melanocytes in a Dose-Dependent Manner, but Does Not Eliminate Dead Cells

To verify whether KGM restored proliferation of cells affected by UVB exposure or induced the growth of new and healthy cells that had not been affected, UVB-irradiated HEMns were cultured for one week with varying concentrations of KGM and stained using senescence-associated β- galactosidase (SA-β-Gal), a common aging marker [34,35], for microscopy visualization (Figure 3A). SA-β-Gal-positive cells were absent in the CTL group but were clearly observed in UVB-exposed cells (Figure 3A, upper panel). In the UVB exposed KGM-treated group, the intercellular spaces created by cell death were gradually filled with newly proliferated non-stained cells (Figure 3A, lower panel). The growth of the new cell population was dependent on KGM concentration (Figure 3A, lower panel). 

To quantitatively assess these results, total cell counts were measured on the 14th day following UVB irradiation. The number of the UVB-exposed KGM untreated cells decreased to about 57% of the number of cells in the CTL group (Figure 3B). However, total cell counts gradually increased with increasing KGM concentrations in the UVB-exposed KGM-treated group. Moreover, there were no significant differences in cell numbers between HEMns treated with 20 μg/mL KGM following UVB exposure and UVB-unexposed CTL cells. In addition, UVB exposure induced the expression of the senescence marker, SA-β-Gal (Figure 3C), whose activity gradually reduced in the presence of KGM in a dose-dependent manner, as the total cell count increased (Figure 3C). The number of cells that stained positive for SA-β-Gal-reached a maximum after UVB exposure and remained constant despite the increase in the KGM concentration, suggesting that KGM had no effect on the removal of SA-β-Gal-stained senescent cells (Figure 3D). As shown in Figure 3E, the change in the proportion of SA-β-Gal-positive cells following KGM treatment after UVB exposure reflected an anti-aging effect of KGM under UVB-induced acute senescence, likely caused by promoting the proliferation of young cells without removing senescent cells.

### 3.3. KGM Treatment Suppresses Hyperpigmentation in UVB-Induced Senescent Human Melanocytes

A previous study reported that UVB irradiation induced senescence and hyperpigmentation in HEMns through the activation of p53-dependent melanogenic-related factors [13]. Therefore, we sought to verify whether hyperpigmentation is regulated by KGM in UVB-induced acute senescence in HEMns.

KGM-dependent physiological changes in senescence were evaluated by measuring melanin accumulation and content level using image analysis and absorbance measurements, respectively (Figure 4A). The pigmentation changes of dissolved melanin strongly increased following UVB exposure compared to CTL, but gradually decreased in a dose-dependent manner in the presence of KGM (Figure 4A, upper panel). Similarly, the melanin content also significantly increased after UVB exposure and decreased upon KGM treatment (Figure 4A, lower panel).

To determine the relationship between the physiological changes and pigmentation-related factors, mRNA expression levels of melanogenesis-related factors, tyrosinase (TYR), tyrosinase- related protein 1 (TRP1), and TRP2 [36], were analyzed after 48 h of cultivation using RT-qPCR (Figure 4B,C). Results showed that the expression levels of pigmentation-related genes were significantly increased by UVB exposure in the untreated group compared to the CTL group, and were gradually suppressed following KGM treatment in a dose-dependent manner (Figure 4B,C). Subsequently, the changes in pigmentation, aging-related factors and degree of DNA damage at day 5 of cultivation with KGM after UVB exposure were analyzed by Western blotting. Specifically, we focused on the expression levels of pigmentation related factors, TYR, TYRP1, and TYRP2, and aging-related factors, p53 together with cyclin dependent kinase inhibitor 1A (CDKN1A; p21 cell cycle-inhibitory protein, p21CIP), and gamma H2AX (γ-H2AX), a DNA damage marker confirmed in a previous study [13]. Overall, the UVB-induced senescent group showed strongly increased expression of aging, pigmentation-related factors, and DNA damage markers compared to the CTL group (Figure 4D,E). Furthermore, the expression of each protein decreased gradually depending on the concentration of KGM, suggesting that UVB-induced senescence and hyperpigmentation improved as cell population increased. Additionally, KGM-treated HEMns promoted the expression of Ki-67, a cell growth marker, inhibited one of senescence associated secretory phenotype (SASP), interleukin 1 beta (IL-1β), and senescence marker, p16^INK4A^, also the known cyclin-dependent kinase inhibitor 2A (CDKN2A), respectively (Appendix A). Notably, an evaluation of changes in pigmentation and melanin content by image comparison and quantitative analysis revealed that KGM had no effect on unexposed normal HEMns (Appendix A). Additionally, there were no changes in the mRNA expression levels of pigmentation-related genes, TYR, TRP1, and TRP2 (Appendix A). 

Taken together, these results suggest that KGM might have the positive effect of anti- pigmentation on skin homeostasis by promoting the cell proliferation in UVB-induced acute senescent HEMns rather than by suppressing pigmentation in normal HEMns.

### 3.4. KGM Reconstitutes UVB-Damaged Human Primary Fibroblasts

After confirming that KGM-dependent cell growth promotion induced anti-pigmentation in UVB-induced accelerated senescence in HEMns, we then investigated whether this restorative function of KGM was universally applicable to human skin cells using HEFs, a key model for UV irradiation-induced cell damage [37,38].

As previously described in the Methods, physiological and biological changes in UVB-irradiated HEFs cultured in KGM-containing media for two weeks were evaluated by analyzing morphological changes using microscopy (Figure 5A, upper panel). UVB-damaged HEFs displayed serious phenotypic changes, including intercellular spaces due to cell elimination. The remaining cells also had abnormal shapes and sizes distinct from those of unaffected cells. Cultivation with media containing KGM refilled the empty spaces with new HEFs depending on KGM concentration (Figure 5A, lower panel). Notably, the effect of KGM on UVB-damaged HEFs mirrored that of KGM on HEMns in the previous experiment.

Quantitative analysis showed that the number of HEFs was reduced by UVB exposure, but gradually increased in a concentration-dependent manner in the presence of KGM (Figure 5B). Additionally, whereas the number of SA-β-Gal-positive cells increased following UVB exposure, the number of cells staining positive for SA-β-Gal remained constant regardless of the increase in KGM concentration (Figure 5C). Moreover, unexposed HEFs cultured with KGM were relatively less damaged than after UVB exposure, likely due to KGM-mediated increase in the total number of cells, even as the number of SA-β-Gal-positive cells was maintained (Figure 5D).

## 4. Discussion

Previous studies have shown that the aging of the human skin is not only inherently caused by internal factors, but also by drastic environmental changes and external factors such as UVB exposure, which strongly impede skin homeostasis [4,5,6,7]. The main challenge of UV-induced skin damage is that the injury is thought to be irreversible. Thus, it is difficult to restore the skin to its previous state. Due to widespread social and health problems caused by UV exposure, there have been various attempts at commercial product development and pre-emptive requirements for regulation [8,9,10,11]. The present study suggests the possibility of KGM to overcome the sudden disruption of skin homeostasis caused by UV irradiation in HEMns and HEFs, which are particularly sensitive components of the skin to environmental changes.

Despite the various applications of KGM [18,19,20,28,29,30,31], many studies related to the effects of KGM on the human skin have only focused on a sheet- or patch-type polymer of KGM rather than on the properties of the material itself [19,20,21]. There are two main reasons for this. The first is that the unique characteristics of KGM with respect to water absorption or clustering may pose a challenge to elucidating its functions as an individual material. The second reason is that the various effects of KGM, including its therapeutic potential, have already been generally demonstrated in studies using a patch or cluster form. In particular, the clinical applications of KGM polymers in improving inflammation after skin damage or acne have been reported [32,33]. Given this background, in the present study, we focused on the biological effectiveness of non-clustered KGM, rather than on physical approaches using conventional patches or sheet polymers.

First, we established the appropriate KGM concentrations for evaluating its effect as a solution rather than as the commercially-available patch or sheet forms. We confirmed that a wide range of KGM concentrations (1–20 μg/mL) was harmless to cell viability, suggesting that KGM could be considered a safe natural material for HEMns. However, as KGM is not readily dissolved in water at concentrations of 3% or higher [22,23], determining the suitable concentrations for manufacturing is necessary since the concentrations used in this study are set in vitro.

The changes in cell morphology and related factors after the exposure of HEMns to UVB were consistent with those of a previous report [13]. In addition, microscopy image analysis showed that the induction of cell growth was dependent on KGM concentration. Consistent with these results, analysis of SA-β-Gal staining and degree of senescence revealed that the number of stained cells was not affected by KGM treatment, even though the cell morphology had been altered. Furthermore, the inability of KGM to clear cells damaged by UVB exposure was a typical challenge that could not be overcome. In previous studies, anti-aging researchers have argued that the elimination of individual aging cells took precedence over the inhibition of aging-related factors. However, our results suggest that culturing UV-exposed cells with KGM results in significant cell regrowth, so that the population appears to recover to the levels prior to UVB exposure, even though in the absence of direct improvement in aging-related factors. Therefore, the overall increase in the cell count could be due to the inhibition of senescence or damage during UVB-induced acute injury. In other words, KGM helps overcome UVB-induced damage by increasing the cell count (without the removal of aging cells) in a concentration-dependent manner.

Acute changes in primary melanocytes due to UVB exposure cause not only external physiological changes such as melanin accumulation but also disrupt the inherent production of melanin by melanocytes [13]. Our results confirmed that UVB exposure caused an increment of melanin synthesis and gene and protein expression of pigmentation-related factors. These changes suggested that KGM inhibits pigmentation after UVB exposure. In this study, the efficacy of KGM in HEMns was validated only under harsh UVB exposure conditions. However, we also confirmed that KGM itself has no direct anti-pigmentation effect under normal conditions. That is, KGM slightly induced the growth of normal HEMns in a concentration-dependent manner (data not shown). Moreover, the normalized cell counts showed that KGM had no effect on the pigmentation-related factors. These results suggest that KGM itself likely does not suppress pigmentation, but could lead to changes in the cellular environment by inducing cell regrowth in UVB-induced acute senescence.

Many previous studies on UVB-mediated senescence or aging-like damage with accompanying physiological changes in the skin have used keratinocytes or dermal fibroblasts rather than melanocytes [8,37,38]. Because UVR-induced physiological changes depend on wavelength, skin penetration depth, and the outermost position of the skin layer, we focused on HEMns, which are located near the basal layer in the innermost layer of the skin [11,12,13]. In addition, there is a lack of clear criteria for skin senescence or aging in keratinocytes or fibroblasts. Therefore, these results must be interpreted carefully since the detailed mechanism of human skin aging after UV exposure remains to be fully elucidated.

As with HEMns, we found that KGM also somewhat improved the overall physiological changes to HEFs caused by UVB exposure, and improved the SA-β-Gal staining ratio. However, SA- β-Gal activity by itself is not a sufficient marker of aging. Therefore, although it has been applied in several studies on aging and aging-related factors, the interpretations and contexts are limited and need careful considerations [13,39] even though the mRNA expression level of another senescence marker, IL-1β was investigated. The lack of change in the number of SA-β-Gal-stained cells in addition to a change in the ratio of stained cells suggested that KGM has a general positive effect on the cellular environment. Although whether UVR induces overall senescence or the aging of HEFs remains to be elucidated, the increase in the proportion of stained cells strongly suggests that UVB promoted a senescence-like environment such as aging acceleration, and that KGM might have a reconstitution effect in a concentration-dependent manner.

Taken together, the present results presumably suggest that KGM could transform the skin environment from a UVR-induced acute senescence/damaged cellular state to a normal cellular condition via the promotion of cell growth. However, further detailed experiments are required to investigate the reconstitution effects of KGM on the dermal environment and the underlying mechanism, including identification of a KGM receptor and Sry-related HMG-Box 10 (Sox10), the proliferation-associated gene in HEMns thought to mediate the observed effects [40,41,42]. Unlike conventional anti-aging materials, which are typically designed to inhibit or eliminate aging-related factors in individual cells, KGM appears to have a different mechanism of action by improving the overall cell environment to overcome UVR-induced changes. Combining KGM with other functional substances that downregulate non-functional proteins deposited in skin, such as denatured elastin on the upper dermis of solar elastosis [43,44] might enhance the effectiveness of this material. In conclusion, KGM is an attractive natural material with potential as an anti-aging product, which further expands the industrial value of this beneficial nutrient.

## Figures and Tables

**Figure 1 nutrients-12-02779-f001:**
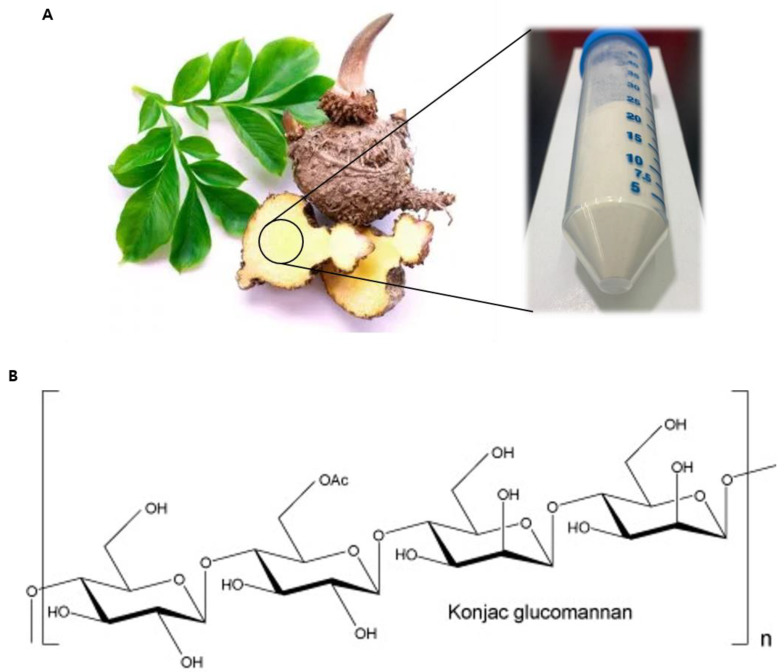
The shape and main composition of konjac plants. (**A**) Image of a konjac plant and the starchy root, which is known as the corm. (**B**) The structure of β-1,4-linked d-glucose and d-mannose repeating units in konjac glucomannan (KGM).

**Figure 2 nutrients-12-02779-f002:**
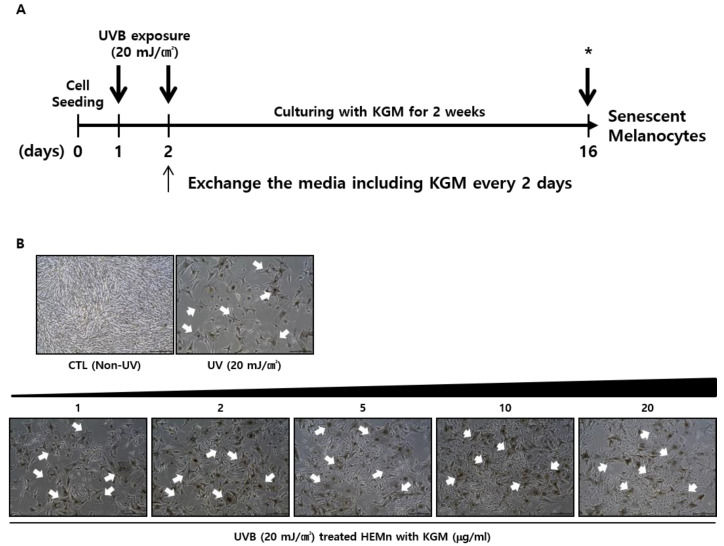
Effects of KGM treatment on UVB-induced senescence in human epidermal primary melanocytes (HEMns). (**A**) Schematic illustration showing the induction of senescence in HEMns by UVB irradiation. The asterisks (*) indicate cell harvesting times for the analysis of senescence after 2 weeks of culture. (**B**) Morphological changes in HEMns (*n* = 3) after UVB irradiation and cultivation with various concentrations of KGM were visualized by microscopy. Senescent HEMns displayed abnormal morphologies following UVB exposure, including flat shapes, darkening, numerous dendrites and enlarged bodies (arrows). Scale bar = 50 μm. CTL, control; UVB, ultraviolet (UV) B; KGM, konjac glucomannan.

**Figure 3 nutrients-12-02779-f003:**
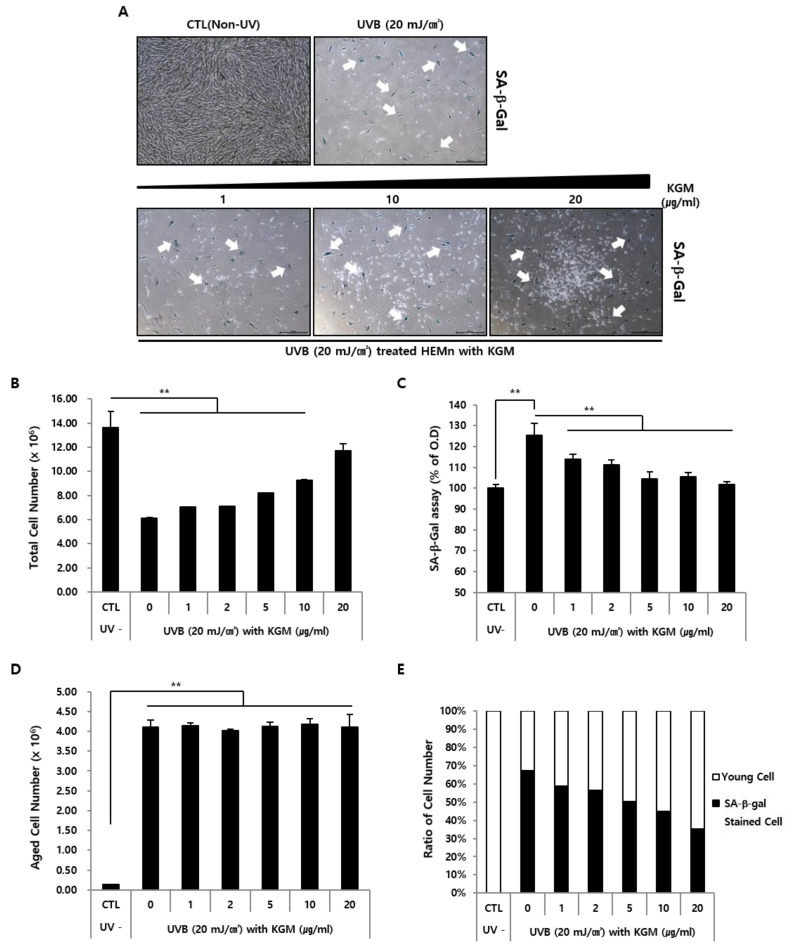
UVB-induced senescence in HEMns indicated by an increase the number of SA- β- Gal- positive cells, and reduced proportion of senescent cells in the presence of KGM. After UVB irradiation, HEMns (*n* = 3) were cultured with various concentrations of KGM for 1 week. (**A**) Senescent HEMns were visualized by microscopy using SA-β-Gal staining (arrow). Scale bar = 500 μm. (**B**) Total numbers of HEMns (*n* = 3) were measured by cell counting. (**C**) Senescence activity (*n* = 3) was determined using the SA-β-Gal assay. The data are presented as the percentage of absorbance values. (**D**) The number of senescent HEMns (*n* = 3) was measured by counting SA- β- Gal- positive cells. The data are presented as mean ± SD (** *p* < 0.01; unpaired Student’s *t*-test). (**E**) Cellular state in SA-β-Gal-stained and unstained HEMns (*n* = 3).

**Figure 4 nutrients-12-02779-f004:**
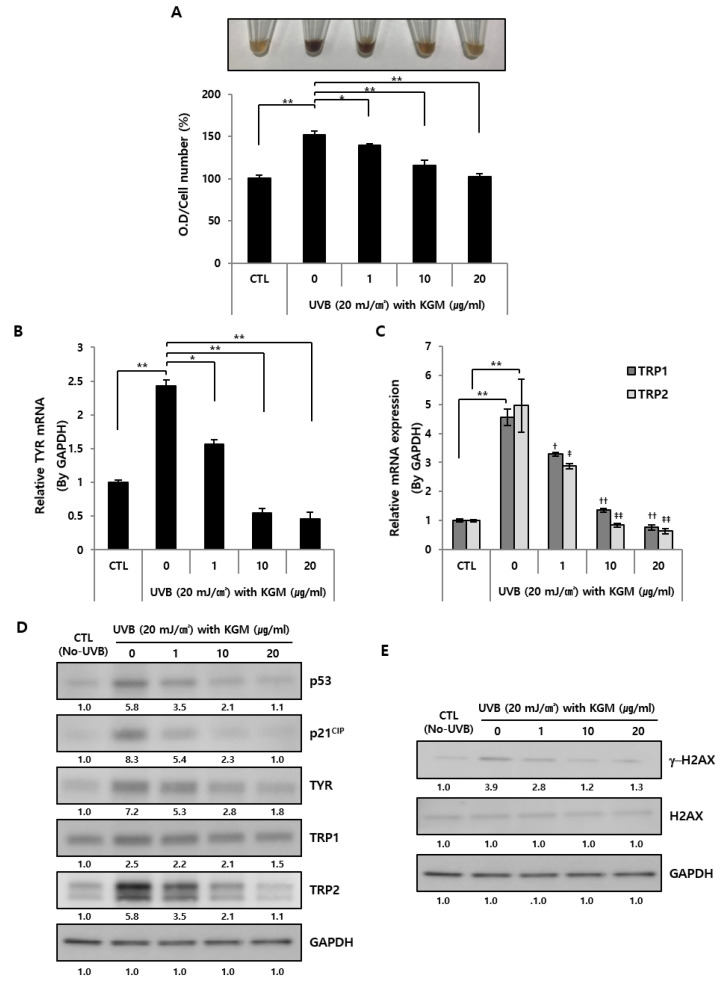
The acceleration of hyperpigmentation in UVB-induced senescent HEMns was regulated by KGM in a dose-dependent manner. The senescence of HEMns was induced by UVB irradiation twice followed by cell culturing with various concentrations of KGM. Proteins and melanins were extracted by lysis from each group of HEMns. (**A**) The melanin content was visualized (upper) or quantified by measuring the absorbance (O.D, optical density) at 450 nm and normalized by total cell count numbers (lower) (*n* = 3). (**B**) mRNA expression levels were determined by Real-time quantitative polymerase chain reaction (RT-qPCR) using specific Taqman probes for tyrosinase (*TYR)**,* (**C**) tyrosinase- related protein 1 (*TRP1*), and *TRP2*. The data are presented as mean ± SD (* *p* < 0.05, ** *p* < 0.01 compared to CTL (*n* = 3): ^†/‡^
*p* < 0.05, ^††/‡‡^
*p* < 0.01 compared to 0 group (*n* = 3) [untreated after UVB exposure] for each gene; unpaired Student’s *t*-test). (**D**,**E**) Protein expression levels were analyzed by Western blotting using specific antibodies for aging-related factors, p53 and p21CIP, and pigmentation-related factors, TYR, TRP1, and TRP2 and DNA damage marker, gamma-H2A histone family member X (γ-H2AX). Glyceraldehyde 3-phosphate dehydrogenase (GAPDH) was used as a control (*n* = 3).

**Figure 5 nutrients-12-02779-f005:**
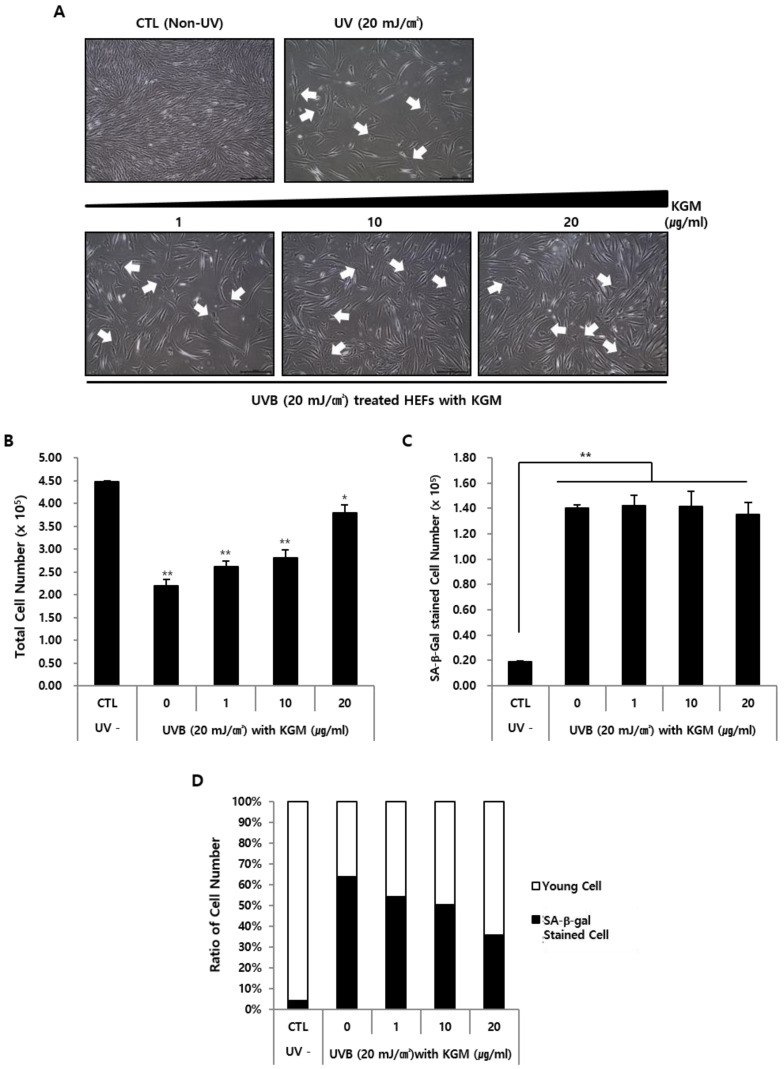
UVB-induced damage in human embryonic fibroblasts (HEFs) displaying abnormal shapes, and improvement of the overall cellular state by KGM. After UVB irradiation twice, HEFs were cultured with various concentrations of KGM for an additional 2 weeks. (**A**) Morphological changes of UVB-exposed HEFs (*n* = 3) were visualized using microscopy. Arrows indicate the abnormal cell shape. Scale bar = 200 μm. (**B**) The change in total number of HEFs and (**C**) SA-β-Gal-positive HEFs was measured by cell counting. The data are presented as mean ± SD (* *p* < 0.05, ** *p* < 0.01; unpaired Student’s *t*-test) (*n* = 3). (**D**) The cellular state was analyzed by comparing SA-β-Gal-stained HEFs (*n* = 3) and unstained HEFs (*n* = 3).

**Table 1 nutrients-12-02779-t001:** Specifications of the konjac glucomannan (KGM) used in this study.

Category	Viscosity (mPa.s)	Glucomannan (%) ≥	pH (1% in DW)	Water (%) ≤	Ash (%) ≤	Lead (Pb) (mg/kg) ≤	Sulfate (g/kg) ≤	Arsenic (As) (mg/kg) ≤	Aflatoxin B1 (ug/kg) ≤
Purified KGM	36,000	95	5.0~7.0	8	2	0.8	0.004	2	5

DW, distilled water.

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
