# Peer review of "Effect of Konjac Glucomannan (KGM) on the Reconstitution of the Dermal Environment against UVB-Induced Condition"

_nutrients, 2020, doi:10.3390/nu12092779_

Round 1

Reviewer 1 Report

The introduction requires more coherence, an improved discussion of the model, and the senescent phenotype. Are your papers the only ones describing this phenomenon? Can you detail it in other cell types? Some sentences don't make sense. For example, the first sentence is poorly worded. Also what is a hyper-dermis? Do you mean dermis or hypodermis? Neither makes sense in the context of the study anyway, when considering you are mainly looking at melanocytes in this study, which are present in the epidermis. Are your UV classifications correct? The last paragraph is extremely repetitive.

Some of the details in the methods need improving, especially in describing how the data is presented. For example, in section 2.3 it is stated that data is presented as the percentage of absorbance value. However, in Figure S1 it is actually presented as  cell viability (%). Section 2.4 describe how the data is presented for the absorbance assay and re-word the sentence for the data output for the staining assay.

Results.

  1. You don't mention how many n's have been performed. This needs to go into each figure legend.
  2. I suggest you look at how often you have used the word 'validate', and whether it is used correctly when you have. 'Investigated' might be a better word to use. Also you use of the words 'such as' is incorrect. Such as is not required if you are listing everything you are measuring.
  3. why do you have photos of tubes in Figure 4. Doesn't the absorbance data show this exact thing quantitatively?
  4. why do you measure the effect of KGM on viability at 24 to 72 hours when the cells are incubated for 2 weeks with KGM?
  5.   section 3.3. I don't understand your reasoning and interpretation in this section.

Discussion

A lot of space is spent at the beginning of the discussion addressing cluster vs non-cluster forms of KGM and research previously undertaken. This is not discussed in the introduction or methods or even results where there is some discussion of KGM, yet from the discussion it seems a major point driving the study. Should be discussed earlier in the paper if it is indeed important. 

Author Response

The introduction requires more coherence, an improved discussion of the model, and the senescent phenotype. Are your papers the only ones describing this phenomenon? Can you detail it in other cell types? Some sentences don't make sense. For example, the first sentence is poorly worded. Also what is a hyper-dermis? Do you mean dermis or hypodermis? Neither makes sense in the context of the study anyway, when considering you are mainly looking at melanocytes in this study, which are present in the epidermis. Are your UV classifications correct? The last paragraph is extremely repetitive.

Thank you for your opinions. In our paper, we hypothesized that natural products can show a function in maintaining homeostasis of skin environment which was exposed to extrinsic factors. As an example, we suggested that Konjac Glucomannan (KGM), used as a food ingredient, have a novel function in skin homeostasis, which was related to anti-aging or alleviation of skin damage after UVB exposure; our report on KGM has not been published elsewhere yet.

To prove this, we used two types of human cells (HEMns and HEFs) which were proper for a better understanding of macroscopic approaches focusing on dermal cell population instead of microscopic approaches focusing on individual cells; herein, we were interested in population changes in dermal region after UVB exposure. Prior to our experiments, we had already known the following researches. Based on the previous literature reported by Rastogi et al. (J. Nucleic Acids, 2010), we thought that UVB is more appropriate than UVA in senescence because DNA is mainly damaged by UVB. Also, in previous paper by Choi et al. (J. Dermatol. Sci., 2018), we secured data that HEMns was aged by repeated exposures by UVB. Considering these, we used HEMns and HEFs in our paper and attempted to prove the reconstitution property of KGM. We didn’t check KGM effect on other cell types.

We are also sorry for the inconvenience. In our manuscript, there were some typos, mistakes and some sentences poorly written. According to reviewer’s comment, we checked the parts pointed out and revised the manuscript again.

Q1. Some of the details in the methods need improving, especially in describing how the data is presented. For example, in section 2.3 it is stated that data is presented as the percentage of absorbance value. However, in Figure S1 it is actually presented as cell viability (%). Section 2.4 describe how the data is presented for the absorbance assay and re-word the sentence for the data output for the staining assay.

A1. Thank you for your opinion. As mentioned, we revised Figure S1 as follows;

[Revised version]

In addition, we revised the sentences in Materials and Methods as follows; 

[Original version]

[Revised version]

Line 115 in page 3,

Following incubation for 30 min at 37 °C, the absorbance was measured at 405 nm using a Synergy H2 microplate reader (BioTek, Winoosk, VT, USA).

Line 112 in page 3,

Following incubation for 30 min at 37 °C, the absorbance was measured at 405 nm using a Synergy H2 microplate reader (BioTek, Winoosk, VT, USA). The experiment was performed in triplicate, and the data were presented as the percentage of absorbance value.

Figure Legend 3C,

(C) Senescence activity measured using the SA-b-Gal assay and

Figure Legend 3C,

(C) Senescence activity (n = 3) was determined using the SA-b-Gal assay. The data are presented as the percentage of absorbance values.

Results.

Q2. You don't mention how many n's have been performed. This needs to go into each figure legend.

A2. We added the number of repeated experiments in each figure legend of manuscript.

Q3. I suggest you look at how often you have used the word 'validate', and whether it is used correctly when you have. 'Investigated' might be a better word to use. Also you use of the words 'such as' is incorrect. Such as is not required if you are listing everything you are measuring.

A3. Thank you for your comments. We chose more proper word in each sentence as requested, which help broad readers to understand. We focused on the following words such as ‘validate’ and ‘such as’ and revised our manuscript again; honestly, our previous version of manuscript received an academic editing service, however these words were not revised in detail. During the revision process, we carefully checked these words and revised sentences.

Q4. Why do you have photos of tubes in Figure 4. Doesn't the absorbance data show this exact thing quantitatively?

A4. Thank you for your comment. In general, some photos are presented in main figures for a better understanding of data. For example, similar photos of tubes as shown in Figure 4 were presented in previous paper reported by other groups as well as our group (Premei et al., Science, 2015; Park et al., J. Invest. Dermatol., 2017; Choi et al., J. Dermatol. Sci. 2018; Cheng et al., Int. J. Mol. Sci., 2018). As previously reported, we believed that presenting both photos of tubes and melanin content seems to be more understandable to readers in Nutrients Journal. In light of this, we showed both data in Figure 4. If you think this doesn’t help readers understand, however, we can either take the data out or put it in supplement.

Q5. why do you measure the effect of KGM on viability at 24 to 72 hours when the cells are incubated for 2 weeks with KGM?

A5. Thank you for your comment. We exchanged a new medium including KGM every 2 day, and cultured additionally for 2 weeks in each group. In a given experiment, we confirmed that 2 day-treatment of KGM-included medium doesn’t show any significant effect on cell viability (Figure S1A). In this aspect, we concluded that 2 day-incubation (48 hours) was appropriate and we had already presented the process in Figure 2A.

In spite of this, our explanation was not sufficient and some parts were less understandable in some aspects. For a better understanding of procedure, we revised the section 2.2 as follows;  

[Original version]

[Revised version]

Line 93 in page 2,

the PBS was removed immediately and replaced with growth medium, and the exposed cells were cultured for an additional 2 weeks with or without KGM. However, control cells were sustained in the same culture conditions without UVB exposure.

Line 92 in page 2,

Then, the PBS was removed immediately and replaced every 2 days with growth medium with or without KGM. The cells were then cultured for 2 more weeks. Control cells were sustained in the same culture conditions without UVB exposure.

Q6.  section 3.3. I don't understand your reasoning and interpretation in this section.

A6. We investigated how KGM affects HEMns, which was followed by previous paper reported by Choi et al. (J. Dermatol. Sci. 2018). Briefly, this paper reported that the cells were aged by repeated exposure by UVB, showing that cells showed the hyperpigmentation due to a higher amount of melanin content. Similar to the previous paper, we repeatedly exposed the HEMns against UVB and treated KGM. As a result, we confirmed that hyperpigmentation condition becomes alleviated in a KGM-concentration manner (Figure 4A). Also, we confirmed that melanogenesis and senescence-related factors were improved as shown in Figure 4B and 4C. Taken together, our data demonstrated that KGM improved UVB-induced environment and recovered it toward normal condition when young cells became dominant after treatment of KGM. In macroscopic point of view, this phenomenon is quite similar to the phenomenon of returning to a young state. To summarize, the section 3.3 suggested that KGM can improve and recover the environment toward the younger state rather than cellular proliferation, which was supported by our data validated.

Q7. Discussion

A lot of space is spent at the beginning of the discussion addressing cluster vs non-cluster forms of KGM and research previously undertaken. This is not discussed in the introduction or methods or even results where there is some discussion of KGM, yet from the discussion it seems a major point driving the study. Should be discussed earlier in the paper if it is indeed important. 

A7. Thank you for your opinion. As requested, we revised the 2nd paragraph in Discussion as follows;

[Original version]

[Revised version]

2nd paragraph in Discussion,

The majority of previous studies on KGM have focused on its physical effects in cluster form [28–31]. Therefore, the benefits of KGM as a health supplement through oral intake have been extensively reported, and its safety has been verified even with long-term use [18–20]. Despite various uses, studies related to the effects of KGM on the human skin have only focused on a sheet- or patch-type polymer of KGM rather than the properties of the material itself [19–21]. There are two main reasons for this. The first is that the unique characteristics of KGM with respect to water absorption or clustering may pose a challenge in elucidating its functions as an individual material. The second potential reason is that the various effects of KGM, including its therapeutic potential, have already been generally demonstrated in studies using a patch or cluster form. In particular, clinical applications of KGM polymers in improving inflammation after skin damage or acne have been reported [32, 33]. Given this background, in the present study, we focused on the biological effectiveness of non-clustered KGM itself, rather than physical approaches using conventional patches or sheet polymers.

2nd paragraph in Discussion (Line 308 in page 11),

Despite the various applications of KGM [18-20, 28-31], many studies related to the effects of KGM on the human skin have only focused on a sheet- or patch-type polymer of KGM rather than the properties of the material itself [19–21]. There are two main reasons for this. The first is that the unique characteristics of KGM with respect to water absorption or clustering may pose a challenge in elucidating its functions as an individual material. The second potential reason is that the various effects of KGM, including its therapeutic potential, have already been generally demonstrated in studies using a patch or cluster form. In particular, clinical applications of KGM polymers in improving inflammation after skin damage or acne have been reported [32, 33]. Given this background, in the present study, we focused on the biological effectiveness of non-clustered KGM itself, rather than physical approaches using conventional patches or sheet polymers.

Reviewer 2 Report

This manuscript entitled “Effect of konjac glucomannan (KGM) on reprofiling of the dermal environment against UVB-induced condition” by Choi et al., demonstrates that KGM promote the cell proliferation of UV radiated-HEMns and HEFs but not removing senescent cells. It is very interest phenomenon that KGM recovers the cell proliferation of UV radiated-HEMns and HEFs.  My comments are written below.

Major comments

Why do you use the ppm as concentration units? I think mg/ml or mM are easier to understand.

UV radiation induces p53 and p21 activation in HEMns, which usually cause G1 or G2/M cell cycle arrest. Did you check the cell cycle arrest of UV radiated-HEMns? In addition, can KGM cancel the cell cycle arrest of them? Because KGM promoted the cell growth of UV radiated-HEMns.

In general, UV radiation causes DNA damage (single stranded break) in the cells. So, did you examine DNA damage UV radiated-HEMns? Can KGM promote DNA repair in HEMns or HEFs after UV radiation?

What is a specific receptor for KGM? Is the receptor already elucidated?

Minor comments

P3, L112, 37°0 should be changed to 37°C.

P3, L117, Waco should be changed to Wako.

Author Response

This manuscript entitled “Effect of konjac glucomannan (KGM) on reprofiling of the dermal environment against UVB-induced condition” by Choi et al., demonstrates that KGM promote the cell proliferation of UV radiated-HEMns and HEFs but not removing senescent cells. It is very interest phenomenon that KGM recovers the cell proliferation of UV radiated-HEMns and HEFs.  My comments are written below.

 Major comments

Q1. Why do you use the ppm as concentration units? I think mg/ml or mM are easier to understand.

A1. Thank you for your comment. We revised the unit of ppm to μg/mL, which is easier to understand as you mentioned.

Q2. UV radiation induces p53 and p21 activation in HEMns, which usually cause G1 or G2/M cell cycle arrest. Did you check the cell cycle arrest of UV radiated-HEMns? In addition, can KGM cancel the cell cycle arrest of them? Because KGM promoted the cell growth of UV radiated-HEMns.

A2. Thank you for your valuable comment. As requested, we’d like to investigate whether KGM affects the cell cycle arrest. However, we couldn’t perform the experiment due to COVID-19 issues; we couldn’t access to other institutes with the instrument (e.g., flow cytometry). We discussed with Dr. Choi SY, a major researcher in previous paper (J. Dermatol. Sci. 2018), about this instead. As a result, we received her positive opinion that KGM could improve the condition of cell cycle arrest by repeated exposure of UVB. Whatever the reason, we apology for not doing additional experiment.

Q3. In general, UV radiation causes DNA damage (single stranded break) in the cells. So, did you examine DNA damage UV radiated-HEMns? Can KGM promote DNA repair in HEMns or HEFs after UV radiation?

A3. Thank you for your comment. As requested, we investigated the expression level of g-H2AX in HEMns, which is a well-known damage marker of DNA (Mah et al. Leukemia. 2010; Siddiqui et al. Mutat. Res. Rev. Mutat. Res., 2015). Based on cell population, the expression level of DNA damage marker was reduced in a KGM-concentration dependent manner, comparing to the control group (the repeated exposed group by UVB). We added data and revised as follows (Figure 4E and section 3. 3)

[Original version]

[Revised version]

Line 244-250 in page 7,

The changes in pigmentation and aging-related factors were subsequently validated by protein expression using western blot analysis at day 5 of cultivation with KGM after UVB exposure. In particular, we focused on pigmentation related factors, TYR, TYRP1, and TYRP2, and aging-related factors, p53 and cyclin dependent kinase inhibitor 1A (CDKN1A; p21 cell cycle-inhibitory protein, p21CIP), which was confirmed in a previous study [13]. Overall, the UVB-induced senescent group showed that the expression level of aging and pigmentation-related factors strongly increased compared to the CTL group (Figure 4D).

Line 237-244 in page 7,

Subsequently, the changes in pigmentation, aging-related factors and degree of DNA damage at day 5 of cultivation with KGM after UVB exposure were analyzed by western blotting. Specifically, we focused on the expression levels of pigmentation related factors, TYR, TYRP1, and TYRP2, and aging-related factors, p53 and cyclin dependent kinase inhibitor 1A (CDKN1A; p21 cell cycle-inhibitory protein, p21CIP), and gamma H2AX (g-H2AX), a DNA damage marker confirmed in a previous study [13]. Overall, the UVB-induced senescent group showed strongly increased expression of aging, pigmentation-related factors and DNA damage markers compared to the CTL group (Figure 4D and 4E).

[Figure 4E in Revised version]

Q4. What is a specific receptor for KGM? Is the receptor already elucidated?

A4. According to previous literatures, mannose receptor (MR, CD206) is a possible receptor for KGM and this receptor is expressed in human dermal cell lines such as fibroblasts as well as keratinocytes (Szolnoky et al. (2001) & Sheikh et al. (2000) [40, 41]). However, it was not reported that this receptor is expressed in human primary melanocytes. We also investigated that MR-related gene and protein is expressed in HEMns, however, we couldn’t confirm the expression, experimentally. In previous literatures, it is reported that MR is existed on the surface of macrophage although we couldn’t test it; whatever the reason, we couldn’t confirm the existence of receptor experimentally. Therefore, we noted that further studies related to this receptor were required in discussion of the manuscript.

Minor comments

Q5. P3, L112, 37°0 should be changed to 37°C.

A5. Thank you for your comment. There might be a typo. We revised it as follows;

[Original version]

[Revised version]

Line 112 in page 3,

Following incubation for 30 min at 37 °0,

Line 111 in page 3,

Following incubation for 30 min at 37 °C,

Q6. P3, L117, Waco should be changed to Wako.

A6. Thank you for your comment. There was a typo. We revised it as follows;

[Original version]

[Revised version]

Line 117 in page 3,

4% formaldehyde solution (Waco, Kyoto, Japan)

Line 117 in page 3,

4% formaldehyde solution (Wako, Kyoto, Japan)

Reviewer 3 Report

In general:

If this manuscript aims to propose that UVB is the pivotal solar environmental factor to induce cell senescence and skin aging, and KGM is an effective natural agent to recover of damaged skin tissue, the repair effect of KGM on DNA damage induced by UVB should be included.  Further, solar UVA is one of the main factors inducing cellular damage leading to cell senescence, it is required to describe the effect of KGM on UVA-induced cell damages, at least in Discussion. Anti-oxidant effect of KGM should be added in the text.

Comments:

  1. How many days are required for the cultured cells exposed to twice UVB radiation to induce senescence, in relevance with different UVB dose irradiated, to proliferate with addition of KGM ?
  2. How many cells showed senescence markers in the experimental condition in which you could induced cell proliferation in the present study? The ratio of cells with and without senescence makers?
  3. In addition to β-gal, other senescence maker should be included in the study to definitely show that non-senescent cells in your experiment increased by KGM treatment.
  4. To clarify the effect of KGM to promote cell proliferation, marker of cell growth, such as Ki-67, should be used.
  5. To study the effect of KGM on reactive oxygen species, the study should add the recovering effect of KGM on H2O2-induced senescent cells.
  6. The anti-melanogenic effect of KGM should be studied using melanocytes stimulated melanogenesis by non-UV light.
  7. Since accumulation of non-functional proteins is considered to be one of the main cause of aging of many tissues, it is better for the manuscript to propose that combined application of KGM with materials which effectively remove non-functional proteins deposited in skin, such as denatured elastin seen at the upper dermis of solar elastosis.

Author Response

In general: If this manuscript aims to propose that UVB is the pivotal solar environmental factor to induce cell senescence and skin aging, and KGM is an effective natural agent to recover of damaged skin tissue, the repair effect of KGM on DNA damage induced by UVB should be included. Further, solar UVA is one of the main factors inducing cellular damage leading to cell senescence, it is required to describe the effect of KGM on UVA-induced cell damages, at least in Discussion. Anti-oxidant effect of KGM should be added in the text.

Thank you for your comments. According to your opinions, we revised our manuscript point-by-point. 

Comments:

Q1. How many days are required for the cultured cells exposed to twice UVB radiation to induce senescence, in relevance with different UVB dose irradiated, to proliferate with addition of KGM ?

A1. Thank you for your comment. We spent 16 days for obtaining data. Briefly, we exposed the cells one day later after seeding cells. After UVB exposure, we exchanged cell culture media including or excluding KGM every 2 day. Then we additionally incubated cells for 2 weeks. As a result, we obtained data.

In addition, the previous paper reported by Choi et al. (J. Dermatol. Sci., 2018) demonstrated that the UVB dose (20 mJ/cm2) was optimized in a given experiment using HEMns. In light of this, we also couldn’t do additional experiment for optimizing UVB dose because we focused on investigating the effect of KGM in HEMns with senescence; we used the same cells and instrument for UVB irradiation.

Q2. How many cells showed senescence markers in the experimental condition in which you could induced cell proliferation in the present study? The ratio of cells with and without senescence makers?

A2. After repeated exposure of UVB, we checked that the stained cells by SA-b-Gal, a representative senescence marker, and confirmed the following number of cells: 4.19 x 10^6 ± 5.12 x 10^4 cell/well (Average ± Standard deviation). The result was presented in Figure 3D.

In Figure 3E, we presented the total ratio of cells after counting SA-b-Gal positive and negative cells.

Q3. In addition to β-gal, other senescence maker should be included in the study to definitely show that non-senescent cells in your experiment increased by KGM treatment.

A3. Thank you for your comment. According to your opinion, we investigated the expression level of interleukin-1 beta (IL-1b), one of the member of senescence associated secretory phenotype (SASP), which was analyzed by RT-qPCR. As a result, we found that the level was drastically increased by repeated UVB exposure, whereas it was statistically reduced in a KGM-concentration dependent manner. We added this result in Supplementary Figure 2B and revised sentences as follows;

[Additional data in Figure S2A]

[Original version]

[Revised version]

Line 253 in page 7,

Furthermore, depending on the concentration of KGM, the expression of each protein level decreased gradually, suggesting that UVB-induced senescent and hyperpigmentation conditions improved by increasing the cell population. Notably, KGM did not demonstrate any effect on UVB unexposed normal HEMn, according to the results of the change in pigmentation and melanin content through image comparison and quantitative analysis, respectively (Figure S2B).

Line 247 in page 7,

Furthermore, depending on the concentration of KGM, the expression of each protein level decreased gradually, suggesting that UVB-induced senescent and hyperpigmentation conditions improved by increasing the cell population. Additionally, KGM-treated HEMns promoted the expression of Ki-67, a cell growth marker, and inhibited one of senescence associated secretory phenotype (SASP), interleukin 1 beta (IL-1b), respectively (Figure S2A and S2B). Notably, an evaluation of changes in pigmentation and melanin content by image comparison and quantitative analysis revealed that KGM had no effect on unexposed normal HEMns (Figure S2C).

Q4. To clarify the effect of KGM to promote cell proliferation, marker of cell growth, such as Ki-67, should be used.

A4. Thank you for your valuable comment. We agree with your opinion. We performed the experiment to clarify the effect of KGM. As a result, we confirmed that Ki-67, a marker of cell growth, was increased in a KGM-concentration dependent manner. We added the data in Figure S2A and revised sentences in the manuscript.

[Additional data in Figure S2A]

[Original version]

[Revised version]

Line 253 in page 7,

Furthermore, depending on the concentration of KGM, the expression of each protein level decreased gradually, suggesting that UVB-induced senescent and hyperpigmentation conditions improved by increasing the cell population. Notably, KGM did not demonstrate any effect on UVB unexposed normal HEMn, according to the results of the change in pigmentation and melanin content through image comparison and quantitative analysis, respectively (Figure S2B).

Line 247 in page 7,

Furthermore, depending on the concentration of KGM, the expression of each protein level decreased gradually, suggesting that UVB-induced senescent and hyperpigmentation conditions improved by increasing the cell population. Additionally, KGM-treated HEMns promoted the expression of Ki-67, a cell growth marker, and inhibited one of senescence associated secretory phenotype (SASP), interleukin 1 beta (IL-1b), respectively (Figure S2A and S2B). Notably, an evaluation of changes in pigmentation and melanin content by image comparison and quantitative analysis revealed that KGM had no effect on unexposed normal HEMns (Figure S2C).

Q5. To study the effect of KGM on reactive oxygen species, the study should add the recovering effect of KGM on H2O2-induced senescent cells.

A5. Thank you for your comment. As requested, we performed an additional experiment using HEMns induced by H2O2. Briefly, we first treated 0.5 mM of H2O2 to HEMns for 24 hours. Then, we added KGM for 48 hours in a KGM-concentration dependent manner. As a result, we confirmed that the level of ROS was reduced depending on the concentration of KGM, which was added as follows;

Q6. The anti-melanogenic effect of KGM should be studied using melanocytes stimulated melanogenesis by non-UV light.

A4. Thank you for your comments. We also checked anti-melanogenic effect of KGM on PMA-induced HEMns, which is normally stimulated melanogenesis condition. As a result, there were no significant anti-melanogenic effect of KGM and these data was added in Figure S2C and S2D.

Q7. Since accumulation of non-functional proteins is considered to be one of the main cause of aging of many tissues, it is better for the manuscript to propose that combined application of KGM with materials which effectively remove non-functional proteins deposited in skin, such as denatured elastin seen at the upper dermis of solar elastosis.

A7. Thank you for your comment. According to reviewer’s opinion, we added the following sentences in Discussion for a better understanding.

[Original version]

[Revised version]

Line 390 in page 12

Unlike conventional anti-aging materials, which are typically designed to inhibit or eliminate aging-related factors in individual cells, KGM appears to have a different mechanism of action by improving the overall cell environment to overcome UVR-induced changes. Therefore, KGM is an attractive natural material for future research as an anti-aging product, which could further increase the industrial value of this beneficial nutrient.

Line 376 in page 12,

Unlike conventional anti-aging materials, which are typically designed to inhibit or eliminate aging-related factors in individual cells, KGM appears to have a different mechanism of action by improving the overall cell environment to overcome UVR-induced changes. Combining KGM with other functional substances that downregulate non-functional proteins deposited in skin, such as denatured elastin on the upper dermis of solar elastosis [43-44] might enhance the effectiveness of this material. In conclusion, KGM is an attractive natural material with potential as an anti-aging product, which further expands the industrial value of this beneficial nutrient.

Reviewer 4 Report

Here, Choi et al describe the effects of Konjac Glucomannan (KGM) in protecting melanocytes from UVB damage. Overall, they determine that KGM is protective and promotes regrowth of non-senescent cells, suggesting this additive may be of benefit to inhibiting hyperpigmentation due to sun exposure. Overall, this work is interesting and of potential utility although multiple critical issues must be addressed below.

  • The use of English throughout is insufficient for publication. Multiple native English speakers need to edit this work, as the majority of the writing throughout is unclear and unprofessional. Key issues include:
    • Use of the term “re-profile” throughout. This is not a specific scientific term and should be replaced with something more descriptive.
    • “twice UVB exposure” does not make sense in English. “Two rounds of UVB exposure” or “the cells were exposed to UVB twice” is correct.
    • There are many subject verb agreement errors throughout.
  • In Figure 1, the structure should be shown as a repeating polymer not as a single compound as drawn.
  • Figure 2 and Figure S1 contain duplicate images. Different figures should have different data. Figure S1B should be Figure 2B for simplicity.
  • Lines 307-309 are inaccurate. This sentence suggests that all skin aging is due to external factors, which is clearly untrue. This should be written such that UVB and other factors can accelerate aging of the skin.
  • Given a fair amount of literature on KGM, the authors should speculate on a potential mechanism of action for KGM in the Discussion.
  • Lines 383-385 are inaccurate as well. The way this sentence is written suggests that KGM has in vivo activity, which is not shown in this work. This should be clarified to better reflect the content of this manuscript.

Author Response

Here, Choi et al describe the effects of Konjac Glucomannan (KGM) in protecting melanocytes from UVB damage. Overall, they determine that KGM is protective and promotes regrowth of non-senescent cells, suggesting this additive may be of benefit to inhibiting hyperpigmentation due to sun exposure. Overall, this work is interesting and of potential utility although multiple critical issues must be addressed below. The use of English throughout is insufficient for publication. Multiple native English speakers need to edit this work, as the majority of the writing throughout is unclear and unprofessional. Key issues include:

Thank you for your comments and opinions. We revised sentences point-by-by-point and additionally received an editing service by Editage. 

Q1. Use of the term “re-profile” throughout. This is not a specific scientific term and should be replaced with something more descriptive.

A1. Thank you for your opinion. As suggested, we agree that ‘re-profile’ seems less scientific in some aspect. Prior to submission, we’d also discussed about this issue in order to find more appropriate word presenting our phenomena. After receiving reviewer’s comment, we deeply discussed about the term. As a result, we concluded that the use of ‘reconstitution’ is better than ‘reprofile’. So we revised it in our manuscript.

Q2. “twice UVB exposure” does not make sense in English. “Two rounds of UVB exposure” or “the cells were exposed to UVB twice” is correct.

A2. According to reviewer’s comment, we revised in the manuscript.

Q3. There are many subject verb agreement errors throughout.

A3. Sorry for the inconvenience. Prior to submission, we received an editing service from professional expert. However, it was not sufficient and there were still typos and mistakes. Therefore, we received an editing service again and resubmitted; the certificate is attached as follows;

Q4. In Figure 1, the structure should be shown as a repeating polymer not as a single compound as drawn.

A4. Thank you for your comment. As requested, we added the structure of a repeated polymer not a single compound as follows;

[Revised version]

Q5. Figure 2 and Figure S1 contain duplicate images. Different figures should have different data. Figure S1B should be Figure 2B for simplicity.

A5. Thank you for your comment. According to reviewer’s comment, we removed Figure S1B and merged it as Figure 2.

[Revised version]

Q6. Lines 307-309 are inaccurate. This sentence suggests that all skin aging is due to external factors, which is clearly untrue. This should be written such that UVB and other factors can accelerate aging of the skin.

A6. Thank you for your comment. There was a mistake. As requested, we revised sentences as follows;

[Original version]

[Revised version]

Line 307 in page 11,

Previous studies have shown that aging of the human skin is not inherently caused by internal factors, but rather drastic environmental changes strongly impede skin homeostasis due to external factors such as UV exposure [4–7].

Line 299 in page 10,

Previous studies have shown that the aging of the human skin is not only inherently caused by internal factors, but also by drastic environmental changes strongly impede skin homeostasis due to external factors such as UV exposure [4–7].

Q7. Given a fair amount of literature on KGM, the authors should speculate on a potential mechanism of action for KGM in the Discussion.

A7. Thank you for your comment. We thought that there is an interaction between KGM and mannose receptor (Szolnoky et al. (2001) & Sheikh et al. (2000)) [40, 41]). In light of this, we speculated that KGM interacts with mannose receptor, presumably triggering some signal cascade although we couldn’t prove a direct data yet. Therefore, we speculated that mannose receptor-related pathway might stimulate a melanocytes proliferation factor such as Sry-related HMG-Box 10 (Sox10), presumably leading to cell proliferation. We added these parts in manuscript as follows; 

[Original version]

[Revised version]

Line 386 in page 12,

Nevertheless, further detailed experiments are required to validate investigate the reconstitute effects of KGM on the dermal environment and the underlying mechanism, including identification of receptors and associated signaling pathways involved in the observed effects.

Line 370 in page 12,

However, further detailed experiments are required to investigate the reconstitution effects of KGM on the dermal environment and the underlying mechanism, including identification of a KGM receptor and Sry-related HMG-Box 10 (Sox10), the proliferation-associated gene in HEMns thought to mediate the observed effects [40-42].

Q8. Lines 383-385 are inaccurate as well. The way this sentence is written suggests that KGM has in vivo activity, which is not shown in this work. This should be clarified to better reflect the content of this manuscript.

A8. Thank you for your comment. We revised it as follows;

[Original version]

[Revised version]

Line 377 in page 12,

Taken together, the present results suggest that KGM can transform the overall skin environment from a UVR-induced acute senescence/damaged cellular state to a normal cellular condition via the promotion of cell growth.

Line 368 in page 12,

Taken together, the present results presumably suggest that KGM could transform the skin environment from a UVR-induced acute senescence/damaged cellular state to a normal cellular condition via the promotion of cell growth.

Round 2

Reviewer 1 Report

I agree it can be published. The authors have changed the manuscript sufficiently to clarify the points I raised.  I still find the first sentence of the introduction very difficult to understand. What are 'skin cell clusters preserved beyond the dermis layer to form a solid barrier'??

Author Response

A1. Thanks for your comments. As we confirmed that the word was not appropriate according to the reviewer's comments, we revised the it as below.

[Original version]

[Revised version]

Line 37 in page 1,

The human skin, comprising almost 16% of the body, maintains a certain level of physiological and biological function (homeostasis) to ensure that the various skin cell clusters beyond the dermis layer are preserved to form a solid barrier [1-3].

Line 37 in page 1,

The human skin, comprising almost 16% of the body, maintains a certain level of physiological and biological function (homeostasis) to ensure that the various skin cells beyond the dermis layer are preserved to form a solid barrier [1-3].

Reviewer 2 Report

Dear Authors

Thank you for your revised manuscript.

The manuscript has been revised well.

I think this will be acceptable for publication in Nutrients.

Thank you for yours Efforts. 

Author Response

Dear Authors

 Thank you for your revised manuscript.

The manuscript has been revised well.

I think this will be acceptable for publication in Nutrients.

 Thank you for yours Efforts. 

A1. Thanks for reviewers’ comments.

Reviewer 3 Report

The manuscript is well revised according to according to the reviewer’s comments, except one point.  The authors are requested to add one more study on the effect of KGM on UVB-induced cell senescence.  The effect of the expression change of p16, an important marker of senescence, should be added in this study.

Author Response

Reviewer 3

Q1. The manuscript is well revised according to according to the reviewer’s comments, except one point.  The authors are requested to add one more study on the effect of KGM on UVB-induced cell senescence.  The effect of the expression change of p16, an important marker of senescence, should be added in this study.

A1. A3. Thank you for your comment. As requested, we investigated the expression level of p16 in HEMns, which is a well-known senescence marker (Nobori et al. Nature. 1994; Stone et al. Cancer Res. 1995; Rayess et al. Int. J. Cancer. 2012). Based on cell population, the expression level of p16 was also reduced in a KGM-concentration dependent manner, comparing to the control group (the repeated exposed group by UVB). We added data and revised as follows (Figure S2C and section 3. 3)

[Original version]

[Revised version]

Line 246 in page 7,

Additionally, KGM-treated HEMns promoted the expression of Ki-67, a cell growth marker, and inhibited one of senescence associated secretory phenotype (SASP), interleukin 1 beta (IL-1b), respectively (Figure S2A and S2B).

Line 246 in page 7,

Additionally, KGM-treated HEMns promoted the expression of Ki-67, a cell growth marker, and inhibited one of senescence associated secretory phenotype (SASP), interleukin 1 beta (IL-1b), and senescence marker, p16, also known cyclin-dependent kinase inhibitor 2A (CDKN2A), respectively (Figure S2A-S2C).

[Figure S2C in Revised version]
